# OpenReview forum: "Red-Teaming NSFW Image Classifiers as Text-to-Image Safeguards"
_ICLR.cc/2026/Conference — ICLR 2026 Conference Withdrawn Submission_

### Official Review · Reviewer_tUh4 · 2025-10-26

**Soundness:** 2
**Presentation:** 2
**Contribution:** 2
**Rating:** 4
**Confidence:** 4

**Summary:**

The paper presents an automated red-teaming framework to expose vulnerabilities of NSFW image classifiers when facing context shifts. The framework follows an exploration–exploitation paradigm. In the exploration stage, the authors create a 36K NSFW dataset. In the exploitation stage, they fine-tune a LLM rewriter that learns from these failure cases to craft new prompts that further evade detection. Finally, the authors show that fine-tuning classifiers on red-teamed data improves robustness against such attacks.

**Strengths:**

1. The proposed exploration–exploitation pipeline, along with the synthetic NSFW dataset, could be valuable for future research.
2. The paper explores the vulnerabilities of NSFW image classifiers from a new perspective, context shifts—benign visual elements that can hide unsafe content.
3. The proposed method proves effective against commercial T2I models.

**Weaknesses:**

1. The choice of baselines seems weak. The authors should include comparisons with more advanced red-teaming methods like SneakyPrompt[1], ART[2], etc, and human-written red teaming datasets i2p[3], Adversarial Nibbler[4], etc.
2. The paper lacks sufficient discussion on how NSFW content was verified during dataset construction. Were all the images verified automatically or manually? It seems unreasonable to assume that all generated images are inherently NSFW.
3. The idea of fine-tuning an LLM to generate red-teaming prompts against LLMs or T2I models is not novel—multiple previous works like MART[5] or ART[2] already employ supervised, RL or multi-round adversarial LLM setups for this purpose.

[1]Yang, Yuchen, et al. "Sneakyprompt: Jailbreaking text-to-image generative models." 2024 IEEE symposium on security and privacy (SP). IEEE, 2024.

[2]Li, Guanlin, et al. "ART: automatic red-teaming for text-to-image models to protect benign users." Advances in Neural Information Processing Systems 37 (2024): 91184-91219.

[3]https://huggingface.co/datasets/AIML-TUDA/i2p

[4]Quaye, Jessica, et al. "Adversarial nibbler: An open red-teaming method for identifying diverse harms in text-to-image generation." Proceedings of the 2024 ACM Conference on Fairness, Accountability, and Transparency. 2024.

[5]Ge, Suyu, et al. "Mart: Improving llm safety with multi-round automatic red-teaming." arXiv preprint arXiv:2311.07689 (2023).

**Questions:**

1. Since the rewriter model is fine-tuned only on nude and violent scenes, can it generalize to rewriting other NSFW types, such as drug abuse or weapons?
2. In Section 4.1, when constructing the NSFW dataset, how to determine whether the generated images are NSFW or not, through automatic filtering or manual verification?
3. How does the method handle borderline or ambiguous cases, such as artistic nudity or implicit sexual content, during dataset construction?
4. In Figure 12, when testing Nano Banana, why is the word “nude” rewritten as “nuuuude”? Is this due to keyword escaping? If “nude” were simply replaced rather than rewritten through a learning-based approach, would the attack against Nano Banana and other t2i models still be effective?

---

### Official Review · Reviewer_H1be · 2025-10-28

**Soundness:** 3
**Presentation:** 3
**Contribution:** 2
**Rating:** 4
**Confidence:** 4

**Summary:**

The paper studies a concept called context shift in which a NSFW image can be missed by a safety classifier if surrounded by benign content. To test current safety classifiers against this issue, authors propose a new exploration-exploitation red teaming approach. In the exploration stage, authors create a diverse NSFW image dataset eligible for context shift. In the exploitation stage, they use the failures found in the exploration stage to train a LLM that rewrites prompts into stealthy version to evade detection by the classifier. Authors perform experiments to show the effectiveness of the approach against various models.

**Strengths:**

1. The paper is clearly written and is easy to follow.
2. The paper studies an important and timely problem.
3. The paper performs a comprehensive set of experiments.

**Weaknesses:**

1. While the paper studies an important problem, it lacks technical rigor. The novelty of the work is also not significant.
2. It is not clear how it is verified if an image is indeed NSFW based on the image generated by the T2I model. I think a human in the loop approach is necessary here. However, in that case the approach might not be as scalable. I wonder if authors have some ideas on this.
3. Some large scale human verification is needed that are unbiased and preferably done by not the authors of the paper to not bias the outcomes.
4. The training of the rewriting LLM can be expensive perhaps there might be more efficient and clever ways to achieve context shift which might be interesting to explore.
5. The studied baselines are really simple and naive. Authors could have used more sophisticated and better baselines to compare their work to (e.g., using current SOTA red teaming approaches for T2I models).

**Questions:**

It is not clear how it is verified if an image is indeed NSFW based on the image generated by the T2I model. I think a human in the loop approach is necessary here. However, in that case the approach might not be as scalable. I wonder if authors have some ideas on this.

---

### Official Review · Reviewer_vEpg · 2025-10-31

**Soundness:** 3
**Presentation:** 3
**Contribution:** 3
**Rating:** 4
**Confidence:** 4

**Summary:**

This paper examines the robustness of NSFW image classifiers that act as safety mechanisms in T2I systems. It identifies a key vulnerability termed *context shifts*, where the inclusion of benign visual elements can cause unsafe images to be misclassified as safe. To systematically study this issue, the authors propose an automated red-teaming framework based on an exploration–exploitation paradigm. The approach leverages large language models and generative image models to synthesize diverse NSFW datasets and to learn prompt rewriting strategies that expose classifier weaknesses. The study further demonstrates that these vulnerabilities can transfer to commercial T2I and text-to-video systems, highlighting potential real-world safety risks. The authors also explore preliminary mitigation strategies, showing that fine-tuning on red-teamed data can improve classifier robustness.

**Strengths:**

+ The paper addresses a timely and practically important research topic.
+ The experiment is comprehensive, spanning multiple classifier families, black-box evaluation, and transfer across generators and commercial deployments.
+ The paper is generally clear and well organized.

**Weaknesses:**

- Most discoveries are made with one primary generator and one captioner, then transferred to a second generator. This leaves open whether the rewriter has overfit to generator/captioner characteristics (style priors, texture biases). Incorporating real-image benchmarks and mixed real/AI distributions (as in UnsafeBench) and swapping captioners (e.g., BLIP/LLaVA vs. GPT) would better establish external validity.
- Following the former point, the rewriter may face similar issues: it is trained on captions generated by a single powerful MLLM (the paper uses GPT-4o as the captioner). That captioner encodes its own priors about what is salient or benign/unsafe, and those priors may overlap with the decision boundary of the target classifiers (or the deployed safeguards). As a result, the rewriter may be exploiting captioner–classifier shared biases rather than learning generalizable evasive strategies.
- For Step in Section 4.1, the paper assumes that the LLM-generated extensions appended to unsafe seed prompts introduce only benign contextual elements, yet it provides no explicit mechanism to verify this assumption. Without filtering or auditing, some extensions may inadvertently add new unsafe or borderline content, confounding the interpretation of context shifts. In such cases, classifier misclassifications could stem from changes in NSFW intensity rather than the presence of neutral context. Incorporating automated text-level moderation, constrained generation templates, or human annotation to confirm benignity would make the causal link between context and classifier failure more credible.
- In Section 5.3, the paper reports success rates of the jailbreak against commercial T2I systems, but omits comparisons with existing attack frameworks such as SneakyPrompt [1], PGJ [2], or MMA-Diffusion [3]. These works similarly target deployed generative systems and quantify jailbreak success and efficiency under real-world constraints. Without benchmarking against them, it remains unclear whether the proposed LLM-based rewriting offers superior effectiveness or simply reproduces known vulnerabilities under a different pipeline.
- The empirical design does not fully isolate whether benign contextual modifications are the true cause of classifier failures. Stronger controls (e.g., counterfactual tests where only a single benign element varies) would better establish causality and rule out confounds from generator stochasticity or style variation, a known issue in semantic adversarial evaluation [4].

## References
- [1] Yuchen Yang et al. SneakyPrompt: Jailbreaking Text-to-image Generative Models. IEEE S&P 2024.
- [2] Yihao Huang et al. Perception-Guided Jailbreak Against Text-to-Image Models. AAAI, 2025.
- [3] Yijun Yang et al. MMA-Diffusion: MultiModal Attack on Diffusion Models. CVPR 2024.
- [4] Dan Hendrycks et al. Natural Adversarial Examples. CVPR 2021.

**Questions:**

- How do you verify that the LLM-extended prompts truly introduce **benign** rather than additional unsafe content? Did you apply any automated moderation or human auditing to confirm that the added contextual elements do not alter NSFW intensity?
- Given that both dataset creation and the rewriter training rely on a single generator (SDXL) and a single captioner (GPT-4o), how do you ensure that your findings are not specific to these components?

---

### Note · Authors · 2025-11-22

**Comment:**

Dear AC, Reviewers vEpg, H1be, and tUh4,

We have decided to withdraw this submission. We sincerely thank you for your time and constructive feedback that we intend to incorporate in the future iteration. Additionally, we are deeply encouraged by the reviewers' consensus that 1) our paper studies and addresses a timely and important problem underlying T2I systems due to NSFW image classifiers' failure; 2) our paper is well organized and clearly written; 3) our experiments are comprehensive.

Before withdrawing, we would like to provide several clarifications regarding concerns raised in the reviews, as we believe these aspects were adequately addressed in our submission but perhaps overlooked:

* Regarding **Captioner Bias** (Reviewer vEpg): The concern that our rewriter might overfit to a single captioner (GPT-4o) was addressed in our ablation study in **Appendix E.8** (referenced in Sec 5.2). We demonstrated that our method generalizes effectively when using a different MLLM (Gemini) as the captioner.
* Regarding **NSFW Verification** (Reviewers tUh4 & H1be): We provided a detailed verification protocol in **Appendix E.2** (referenced in Sec 5.1). We manually examined over 1,000 sampled images, with a strict consensus protocol between two authors, confirming that >91.9% of the generated images were indeed NSFW. We would like to highlight that outsourcing and upscaling this verification to external labelers (as suggested by Reviewer H1be) can be difficult due to significant ethical concerns, platform restrictions (regarding the exposure of human annotators to graphic NSFW content), and model restrictions (to avoid the spread of NSFW content).

**Reason for Withdrawal**: While we stand by the rigor of our current analysis, we recognize that the reviewers have identified several key areas that require a more substantial revision than the rebuttal period allows. We plan to withdraw to restructure our paper and fully integrate these suggestions, specifically by conducting a more comprehensive benchmarking against both SOTA attacks (e.g., SneakyPrompt, MMA-Diffusion, Perception-Guided Jailbreak) and simple baselines (e.g., misspelling NSFW keywords), implementing stricter checks to ensure LLM-extended prompts remain benign, and providing a deeper analysis of the training costs. We believe that holistically addressing these points will result in a stronger contribution, and we look forward to submitting this improved version to another venue in the near future.

Best regards,

The Authors

**Withdrawal Confirmation:**

I have read and agree with the venue's withdrawal policy on behalf of myself and my co-authors.